# Changes in cell morphology and function induced by the *NRAS* Q61R mutation in lymphatic endothelial cells

**Shiho Yasue**, **Michio Ozeki***, **Akifumi Nozawa, Saori Endo, Hidenori Ohnishi**

Department of Pediatrics, Graduate School of Medicine, Gifu University, Gifu, Japan

* ozeki.michio.j5@f.gifu-u.ac.jp

**Data Availability Statement:** All relevant data are within the manuscript and its Supporting information files.

## Abstract

Recently, a low-level somatic mutation in the *NRAS* gene (c.182 A > G, Q61R) was identified in various specimens from patients with kaposiform lymphangiomatosis. However, it is unknown how these low-frequency mutated cells can affect the characterization and surrounding environment of their lesions. To understand the pathogenesis and association of these gene abnormalities, we established *NRAS*^Q61R mutated lymphatic endothelial cells transfected with lentivirus vector and undertook morphological and functional characterization, protein expression profiling, and metabolome analysis. *NRAS*^Q61R human dermal lymphatic endothelial cells showed poor tube formation, a low proliferation rate, and high migration ability, with an increase in the ratio of mutated cells. An analysis of signaling pathways showed inactivation of the PIK3/AKT/mTOR pathway and hyperactivation of the RAS/MAPK/ERK pathway, which was improved by MAPK kinase (MEK) inhibitor treatment. This study shows the theoretical circumstances induced *in vitro* by *NRAS*^Q61R-mutated cells in the affected lesions of kaposiform lymphangiomatosis patients.

## Introduction

Generalized lymphatic anomaly (GLA) and kaposiform lymphangiomatosis (KLA), known as complex lymphatic anomalies (CLAs), are rare multiorgan diseases that cause pleural effusion, pericardial effusion, ascites, and osteolysis due to the proliferation of abnormal lymphatic tissue [1]. In the updated International Society for the Study of Vascular Anomalies 2018 classification, GLA is classified as a lymphatic malformation (LM) [2]. KLA is characterized by spindle-shaped lymphatic endothelial cells (LECs) and presents with more serious symptoms (abnormal coagulation, thrombocytopenia, and hemorrhagic pericardial and pleural effusion) and a worse prognosis than GLA [3]. However, the etiology and pathogenesis of KLA remain to be clarified.

Although genetic analysis has identified low-level somatic mutations in various specimens of patients with vascular anomalies, little has been reported on the associated genetic abnormalities in KLA. Nonetheless, mutations in genes encoding components of the PIK3/AKT/mTOR and RAS/MAPK/ERK pathways have recently been identified in the affected lesions of CLA patients. A somatic mutation in *NRAS* (c.182 A > G, Q61R) was detected in 10 of 11 tested lesions from KLA patients [4]. Our group also detected an *NRAS* mutation in cell-free

**Funding:** The authors received no specific funding for this work.

**Competing interests:** The authors have declared that no competing interests exist.

DNA in the plasma and pleural effusion from KLA patients [5]. The frequencies of the mutated alleles in the affected cells in these reports were very low (lower than 30% or only in a few cells). However, it is still unknown how the low-frequency mutated cells can affect the characteristics and surrounding environment of their lesions.

Animal models and patient-derived cells have been used to investigate the pathogenesis and association of these gene abnormalities [6–8]. Studies in mouse models have provided important findings regarding the mechanisms of *PIK3CA* mutations, which affect the phenotype of LECs and vessel overgrowth. LM patient-derived LECs with mutant *PIK3CA* showed increased proliferation and resistance to stimuli that cause cell death [9]. Mutated *PIK3CA*-expressing murine LECs also showed increased migration and sprouting [10]. Thus, the development of LMs is triggered by a somatic activating *PIK3CA* mutation in LECs, leading to cell autonomous proliferation and migration caused by the activation of PIK3/AKT signaling. However, most LM cells with abnormal gross and microscopic morphology in affected lesions do not have any genetic abnormalities. Therefore, little is known about how the very small proportion of mutated LECs can affect other non-mutated cells or the surrounding environment.

In this study, we established *NRAS*^Q61R-mutated LECs transfected with a lentivirus vector and performed morphological and functional characterizations, protein expression profiling, and metabolome analysis. To elucidate the influence of the mutated LECs on wild-type LECs, we studied the effects of mixing these cell populations. Furthermore, we examined the response of these cells to mTOR and MAPK kinase (MEK) inhibitors.

## Materials and methods

### Cell and vector preparation

Juvenile foreskin-derived human dermal lymphatic endothelial cells (HDLECs) were purchased from PromoCell (Heidelberg, Germany). Cells were cultured in Endothelial Cell Growth Basal Medium-2 (EBM2) (Lonza, Basel, Switzerland) with 10% heat-inactivated fetal bovine serum, penicillin (100 units/mL), and streptomycin (100 μg/mL), and propagated at 37°C under 5% $CO_2$.

cDNA of *NRAS*^Q61R (accession number: NM_002524.5) with a 3′ terminal 6 × His-tag was chemically synthesized and cloned into the pFastBac1 vector (Invitrogen, Carlsbad, CA, USA). The cDNA of *NRAS*^Q61R was cloned into the pLVSIN-EF1α-AcGFP-C1 vector (Takara Bio Inc., Otsu, Japan).

### Gene transfection of HDLEC cell line using lentiviruses

The Lenti-X 293 T cell line (Takara Bio Inc.) and pLVSIN-EF1α-AcGFP-C1 vector were used to transfect *NRAS*^Q61R vectors into HDLECs. The Lenti-X 293 T cell line was spread on 10 cm plates at a density of $5 \times 10^6$ cells per plate and cultured in Dulbecco's Modified Eagle Medium (DMEM) for 24 h. The pLVSIN-EF1α-AcGFP-C1 vector (5.5 μg) was added to 7 μL Lentiviral Mix High Titer Packaging Mix (Takara Bio Inc.), 1500 μL serum-free DMEM, and 45 μL Trans IT-293 Transfection Reagent (Takara Bio Inc.). After 15 min, this mixture was added to Lenti-X 293 T cells and incubated at 37°C. After 24 h, the medium was exchanged with DMEM. After 48 h, the culture medium was collected and passed through a 0.45 μm filter (this solution contained recombinant lentiviruses). HDLECs were spread onto six-well plates at a density of $2 \times 10^5$ cells per well. We prepared the recombinant lentiviral solutions at initial concentrations of 1:4 to 1:10. The medium was discarded from each well, and 2 ml of the recombinant lentiviral solutions were added to each well. Polybrene was added to the medium to a final concentration of 4 μg/ml. To select stable GFP-only or C-terminal GFP-fused *NRAS*^Q61R cells, repeated exchange of the medium and the addition of puromycin (final

concentrations of 1.5 µg/m and 0.5 µg/ml, respectively) were performed every 72 h. *NRAS*$^{Q61R}$ insertion in these cells was confirmed by polymerase chain reaction (PCR). The *GFP* gene was transfected into HDLECs using the same method. These transfected cells and *NRAS* wild-type fluorescent cells were used in the scratch assay.

## Insertion of the *NRAS* mutation

Genomic DNA was extracted from cell pellets using the Sepa Gene Kit (EIDIA Co., Ltd., Tokyo, Japan) in accordance with the manufacturer's instructions. DNA extraction was performed using a single extraction method. After extraction, DNA was stored at 4˚C until use.

The following primers were used for SNP analysis: *NRAS*-forward 5′-CAGGTGGTGTTGGGAAAAGC-3′ and *NRAS*-reverse 5′-CTCGCTTAATCTGCTCCCTGT-3′. The sequences were analyzed using a BigDye Terminator v1.1 Cycle Sequencing Kit and an Applied Biosystems 3130xl Genetic Analyzer (Applied Biosystems, Carlsbad, CA, USA).

## Tube formation assay

Cells were plated on thin layers of Geltrex (Thermo Fisher Scientific, Waltham, MA, USA) at a density of $1 \times 10^4$ cells/well in a 96-well plate, cultured in EBM2, and incubated at 37˚C for 20 h. Images were captured automatically every 5 min using time-lapse video microscopy (WSL-1800-B; ATTO Corp., Tokyo, Japan). After 20 h, microscopic images were taken and analyzed by Wimasis Image Analysis (WimTube; Wimasis GmbH Munich, Germany). Tube formation was observed by microscopy (BZ-9000; KEYENCE, Osaka, Japan). To assess the association between the proportion of mutated cells and the function of tube formation, differences in tube formation function were examined in *NRAS* wild type HDLECs (*NRAS*$^{WT}$) (*NRAS*$^{WT}$ 100% and *NRAS*$^{Q61R}$ 0%), *NRAS*$^{Q61R}$ HDLECs (*NRAS*$^{WT}$ 0% and *NRAS*$^{Q61R}$ 100%), and mixed HDLECs (*NRAS*$^{WT}$ 95% and *NRAS*$^{Q61R}$ 5%, and *NRAS*$^{WT}$ 25% and *NRAS*$^{Q61R}$ 75%). To analyze the function of tube formation, the covered area, total nets, and tube length standard deviation in each image were counted and average values within groups were compared. "Covered area" is the area occupied by cellular components and "total nets" is the number of nets (a structure in which two or more loops are connected). At least three fields per well were examined and each experimental condition was tested in triplicate. In addition, 30 ng/ml mTOR inhibitor (rapamycin (sirolimus); Sigma-Aldrich, St. Louis, MO, USA) or 30 ng/ml MEK inhibitor (trametinib; ChemScene, Monmouth Junction, NJ, USA) was added to these cells to examine drug inhibition.

## Scratch assay

*NRAS*$^{WT}$ and *NRAS*$^{Q61R}$ HDLECs were cultured to 90% confluency in 24-well plates. Cell monolayers were scraped with a 200 µl pipette tip to make a wound, and then the cells were cultured in EBM2. Images were captured automatically every 5 min using time-lapse video microscopy (WSL-1800-B; ATTO Corp., Tokyo, Japan). Quantification of wound areas was performed using image analysis software (ImageJ) at 0, 6, 12, 18, and 24 h [11, 12].

## Cell proliferation assay

A Cell Counting Kit-8 assay (CCK-8; Dojindo Laboratories, Kumamoto, Japan) was used to assess the effects of mTOR inhibitor sirolimus or MEK inhibitor trametinib on cell proliferation, in accordance with the manufacturer's protocol. We seeded $1 \times 10^4$ cells/well in 96-well plates and incubated them with sirolimus (40 ng/ml), trametinib (20 ng/ml), or vehicle control (dimethyl sulfoxide) for 48 h. To assess the association between the proportion of mutated

cells and cell proliferation, the differences were examined in *NRAS*[WT], *NRAS*[Q61R], and mixed HDLECs. We then added 10 μl CCK8 solution to each well with 100 μl medium and incubated the cells for 3 h. After incubation, absorbance was measured at 450 nm using a plate reader (Model 680; Bio-Rad Laboratories, Inc., Hercules, CA, USA).

## Analysis of signaling pathways by western blotting and multiplex protein assay

Cells were lysed in cold lysis buffer (CytoBuster Protein Extraction Reagent, Novagen Inc., Madison, WI, USA) in the presence of a protease inhibitor cocktail (Complete Mini, Roche Diagnostics, Mannheim, Germany). Protein samples were separated by sodium dodecyl sulfate polyacrylamide gel electrophoresis and transferred to polyvinylidene fluoride membranes. The membranes were incubated with 5% skimmed milk solution and probed with the indicated primary antibodies at 4˚C overnight. The membranes were then washed and incubated with secondary antibodies for 1 h at room temperature. Protein bands were visualized using ECL Prime Western Blotting Detection Regent (GE Healthcare; Amersham, UK) and a light-capture cooled CCD camera system (ATTO Corp.). Monoclonal antibodies against AKT (2920S) and polyclonal antibodies against phospho-AKT (Ser473) (p-AKT [Ser473]; 9271S) were purchased from Cell Signaling Technology (Boston, MA, USA). Polyclonal antibodies against extracellular signal-regulated kinase 1/2 (ERK1/2; ab196883) and monoclonal antibodies against phospho-ERK1/2 (p-ERK1/2; ab201015) were purchased from Abcam plc. (Cambridge, UK). Protein expression was normalized to GAPDH (sc-2577; Santa Cruz Biotechnology, Santa Cruz, CA, USA).

11-Plex Akt/mTOR Total Protein Magnetic Bead Kit (cat #: 48-612MAG; MILLIPLEX), Akt/mTOR Phosphoprotein 11-plex Magnetic Bead (cat #: 48-611-MAG; MILLIPLEX), and Phospho/Total ERK 2-Plex Magnetic Bead Kit (EMD Millipore Corporation, Billerica, MA, USA) were used to assess the levels of these proteins. Cells were added to six-well plates and cultured until 90% confluency was achieved. To assess the association between the proportion of mutated cells and the levels of these proteins, the differences were examined in *NRAS*[WT], *NRAS*[Q61R], and mixed HDLECs in triplicate at one timepoint using a single plate. The procedure was performed in accordance with the manufacturer's assay protocols (EMD Millipore). A Luminex 200 machine and MILLIPLEX Analyst software were used for data analysis.

## Suspension array

The concentrations of 16 angiogenic and lymphangiogenic factors in the culture medium of *NRAS*[WT] and *NRAS*[Q61R] HDLECs were evaluated using a commercial kit from Luminex. Culture medium without cells was used as the control. The Human Angiogenesis/Growth Factor Magnetic Bead Panel (MILLIPLEX MAP, HAGP1MAG-12K) was performed in duplicate in accordance with the manufacturer's protocol (EMD Millipore).

## Metabolite extraction

Culture medium was aspirated from the dish, and cells were washed twice with 5% mannitol solution (10 ml and 2 ml for the first and second washes, respectively). The cells were then treated with 800 μl methanol and incubated at room temperature for 30 sec to suppress enzymatic activity. Next, 550 μl Milli-Q water containing internal standards (H3304-1002, Human Metabolome Technologies, Inc. (HMT), Tsuruoka, Yamagata, Japan) was added to the cell extract, followed by further incubation at room temperature for 30 sec. The cell extract was then centrifuged at 2,300 ×*g* for 5 min at 4˚C, after which 700 μl of the supernatant was centrifugally filtered through a Millipore 5 kDa cutoff filter (UltrafreeMC-PLHCC, HMT) at 9,100 ×

*g* for 120 min at 4˚C to remove macromolecules. Subsequently, the filtrate was evaporated to dryness under vacuum and reconstituted in 50 μl Milli-Q water for metabolome analysis at HMT.

## Metabolome analysis (C-SCOPE)

Metabolome analysis was conducted in accordance with HMT's *C-SCOPE* package using capillary electrophoresis time-of-flight mass spectrometry (CE-TOFMS) for cation analysis and CE-tandem mass spectrometry (CE-MS/MS) for anion analysis on the basis of the methods described previously [13, 14]. Briefly, CE-TOFMS and CE-MS/MS analyses were performed using an Agilent CE capillary electrophoresis system equipped with an Agilent 6210 time-of-flight mass spectrometer (Agilent Technologies, Inc., Santa Clara, CA, USA) and Agilent 6460 Triple Quadrupole LC/MS (Agilent Technologies), respectively. The systems were controlled by Agilent G2201AA ChemStation software version B.03.01 for CE (Agilent Technologies) and connected by a fused silica capillary (50 μm *i.d.* × 80cm total length) with commercial electrophoresis buffer (H3301-1001 and I3302-1023 for cation and anion analyses, respectively, at HMT) as the electrolyte. The time-of-flight mass spectrometer was scanned from *m/z* 50 to 1,000 [13] and the triple quadrupole mass spectrometer was used to detect compounds in the dynamic MRM mode. Peaks were extracted using MasterHands automatic integration software (Keio University, Tsuruoka, Yamagata, Japan) [15] and MassHunter Quantitative Analysis B.04.00 (Agilent Technologies) to obtain peak information including *m/z*, peak area, and migration time (MT). Signal peaks were annotated in accordance with HMT's metabolite database, based on their *m/z* values and MTs. The peak area of each metabolite was normalized to internal standards, and metabolite concentrations were evaluated by standard curves with three-point calibrations using each standard compound. Hierarchical cluster analysis and principal component analysis [16] were performed using HMT's proprietary MATLAB and R programs, respectively. Detected metabolites were plotted on metabolic pathway maps using VANTED software [17].

## Statistical analysis

Welch's *t*-test was used to compare two groups, and one-way ANOVA was used to compare three or more groups. For comparisons of three or more groups, p-values were calculated by Tukey's multiple comparison with the 0% group or control group as the reference. P-values were considered significant at <0.05. Data were analyzed using EZR (Easy R) software, which is for R [18]. More precisely, it is a modified version of R commander designed to add statistical functions frequently used in biostatistics.

# Results

## *NRAS*^Q61R^ HDLECs are large and have irregular-shaped morphology

Gene transfection of *NRAS*^Q61R^ was confirmed by PCR. *NRAS*^Q61R^ HDLECs showed larger and irregular-shaped morphology compared with *NRAS*^WT^ HDLECs (Fig 1a and 1b) microscopically. Compared with *NRAS*^WT^ HDLECs, *NRAS*^Q61R^ HDLECs were larger, and their shapes were more irregular.

## Poor tube formation in *NRAS*^Q61R^ HDLECs is improved by inhibition of the mTOR/MEK pathway

A tube formation assay revealed that *NRAS*^WT^ HDLECs formed well-regulated lumen structures and tubes, but the sheer-like proliferation patterns and formation of capillary-like

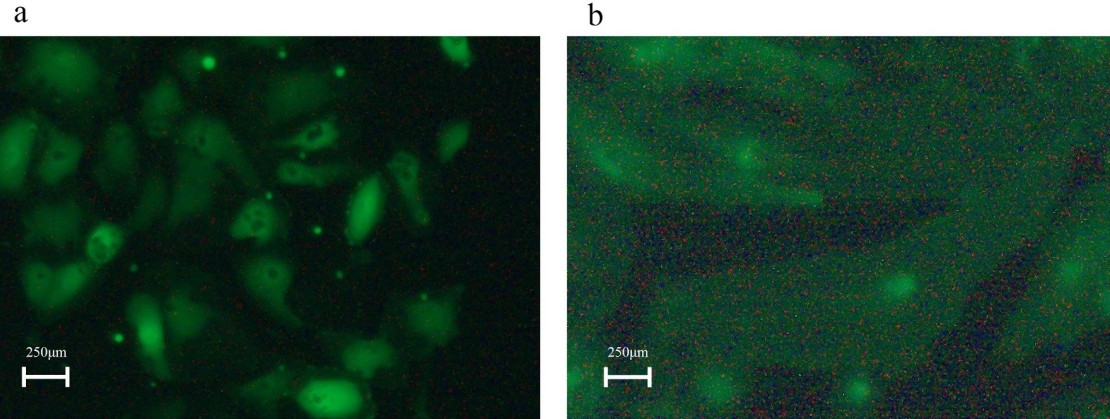

**Fig 1. Morphology of human dermal lymphatic endothelial cells (HDLECs) expressing *NRAS*^WT and *NRAS*^Q61R.** 1a: *NRAS*^WT HDLECs had a uniform and short-spindle-shaped morphology. 1b: *NRAS*^Q61R HDLECs had relatively large and irregular morphology.

structures in $NRAS^{Q61R}$ HDLECs were markedly reduced compared with those of $NRAS^{WT}$ HDLECs (Fig 2A). In the stepwise examination of $NRAS^{Q61R}$ HDLEC mixtures, the covered area, total nets, and tube length standard deviation of each cell line were analyzed. The greater the number of mutated cells, the larger the covered area and the lower the number of total nets (p<0.01) (Fig 2B). The mean values of the covered areas occupying cellular components of the 25% and 100% $NRAS^{Q61R}$ HDLEC mixtures were significantly higher than those of the 0% group (p = 0.030 and p<0.001, respectively) (Fig 2B-a). The number of nets of 0% $NRAS^{Q61R}$ was significantly higher than those of 5%, 25%, and 100% $NRAS^{Q61R}$ (p = 0.034, p<0.001, and p<0.001, respectively) (Fig 2B-b). The standard deviation of tube lengths showed no difference between 0%, 5%, and 25% $NRAS^{Q61R}$, but that of 100% $NRAS^{Q61R}$ was significantly longer than those of 0%, 5%, and 25% $NRAS^{Q61R}$ (p = 0.004) (Fig 2B-c).

Time-lapse photography showed dynamic formational changes in the capillary-like structures of $NRAS^{Q61R}$ HDLECs, and the cell migration was different from that of $NRAS^{WT}$ HDLECs (S1 and S2 files). As the number of mutated cells increased, the impairment of the formation of uniform and continuous loops became greater, which was concentration dependent.

The covered areas occupying cellular components of 100% $NRAS^{Q61R}$ HDLECs treated with sirolimus and trametinib were significantly smaller than those of the controls (p = 0.004 and p = 0.004, respectively) (Fig 2C-a). The number of nets of 100% $NRAS^{Q61R}$ treated with sirolimus and trametinib was significantly higher than that of the controls (p = 0.015 and p = 0.005, respectively) (Fig 2C-b). The standard deviations of tube lengths of 100% $NRAS^{Q61R}$ treated with sirolimus and trametinib were significantly shorter than those of the controls (p = 0.025 and p = 0.025, respectively) (Fig 2C-c). Sirolimus and trametinib treatment improved the poor tube-forming ability resulting from the *NRAS* mutation (Fig 2D-a and b).

## $NRAS^{Q61R}$ HDLECs have a higher migration ability than $NRAS^{WT}$ HDLECs

The scratch assay showed that $NRAS^{Q61R}$ and $NRAS^{WT}$ HDLECs had migrated into the void and completely closed the space. Migration of $NRAS^{Q61R}$ HDLECs was more active than that of $NRAS^{WT}$ HDLECs and the space disappeared within 24 h (Fig 3A). The space between wild-

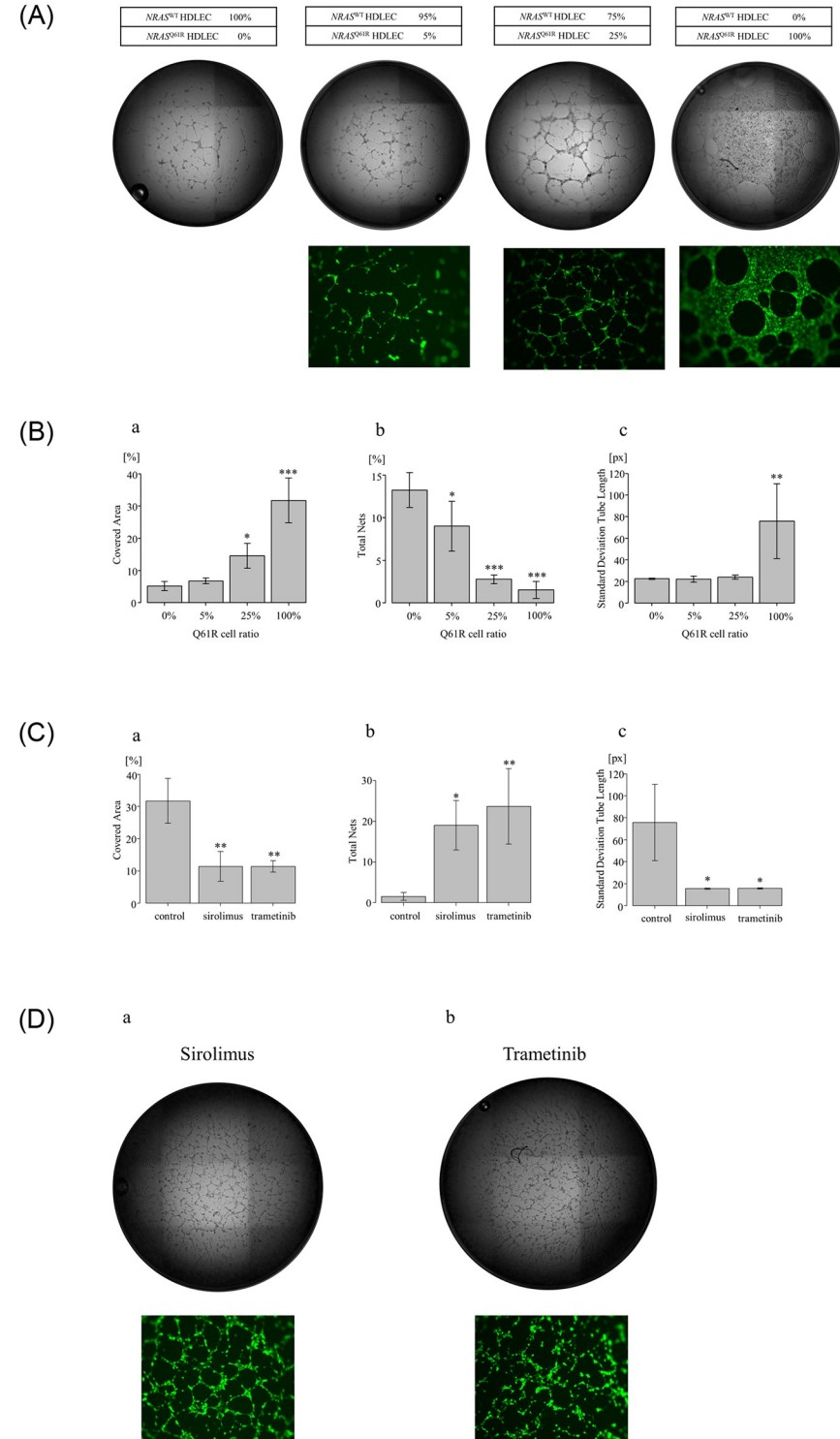

**Fig 2. Tube formation assay of *NRAS*^WT^ and *NRAS*^Q61R^ HDLECs.** 2A. Ratios of *NRAS*^WT^, *NRAS*^Q61R^ and mixed HDLECs, and microscopy images. Cells were mixed at the indicated ratios, and tube formation assays were performed. *NRAS*^WT^ HDLECs showed well-regulated lumen structures and tube formation, but *NRAS*^Q61R^ HDLECs exhibited sheet-like proliferation, and the formation of capillary-like structures was markedly lower than that of *NRAS*^WT^ HDLECs. An increasing proportion of *NRAS*^Q61R^ HDLECs (fluorescent cells) is associated with increasing sheet-like proliferation and decreasing numbers of capillary-like structures. 2B. Stepwise examination of *NRAS*^Q61R^ HDLEC mixtures, the covered area (occupied area by cellular components) (a), total nets (number of nets) (b), and tube length

standard deviation (c), analyzed by WinTube. Bars represent the mean ± SD of triplicate wells. *p<0.05, **p<0.01, ***p<0.001, compared with 0%. 2C. Covered area (a), total nets (b), and tube length standard deviation (c) of *NRAS*$^{Q61R}$ HDLECs with or without treatment. Bars represent the mean ± SD of triplicate wells. *p<0.05, **p<0.01, compared with *NRAS*$^{Q61R}$ HDLECs (no treatment). 2D. Microscopy images of the tube formation of *NRAS*$^{Q61R}$ HDLECs treated with an mTOR inhibitor, sirolimus (30 ng/ml), (a) and the MEK inhibitor trametinib (30 ng/ml) (b).

type cells disappeared after 60 h. The space ratio of *NRAS*$^{Q61R}$ at 0, 6, 12, 18, and 24 h decreased rapidly. However, that of *NRAS*$^{WT}$ HDLECs showed a slow decrease (Fig 3B).

## Increased proportion of mutant cells decreases the cell proliferation rate

The cell proliferation assay of *NRAS*$^{Q61R}$ and *NRAS*$^{WT}$ mixed HDLECs showed that the absorbance of 100% *NRAS*$^{Q61R}$ was significantly lower than that of *NRAS*$^{WT}$ (p = 0.024) (Fig 3C). There was no difference in absorbance between *NRAS*$^{WT}$ and 5% and 25% *NRAS*$^{Q61R}$. After sirolimus and trametinib treatment, the absorbance of *NRAS*$^{Q61R}$ mixtures decreased significantly compared with that of cells with no treatment. There was no difference between the absorbance of 100% *NRAS*$^{Q61R}$ after sirolimus treatment and that of no treatment, but the absorbance of 100% *NRAS*$^{Q61R}$ after trametinib treatment was significantly lower than that of cells with no treatment (Fig 3C).

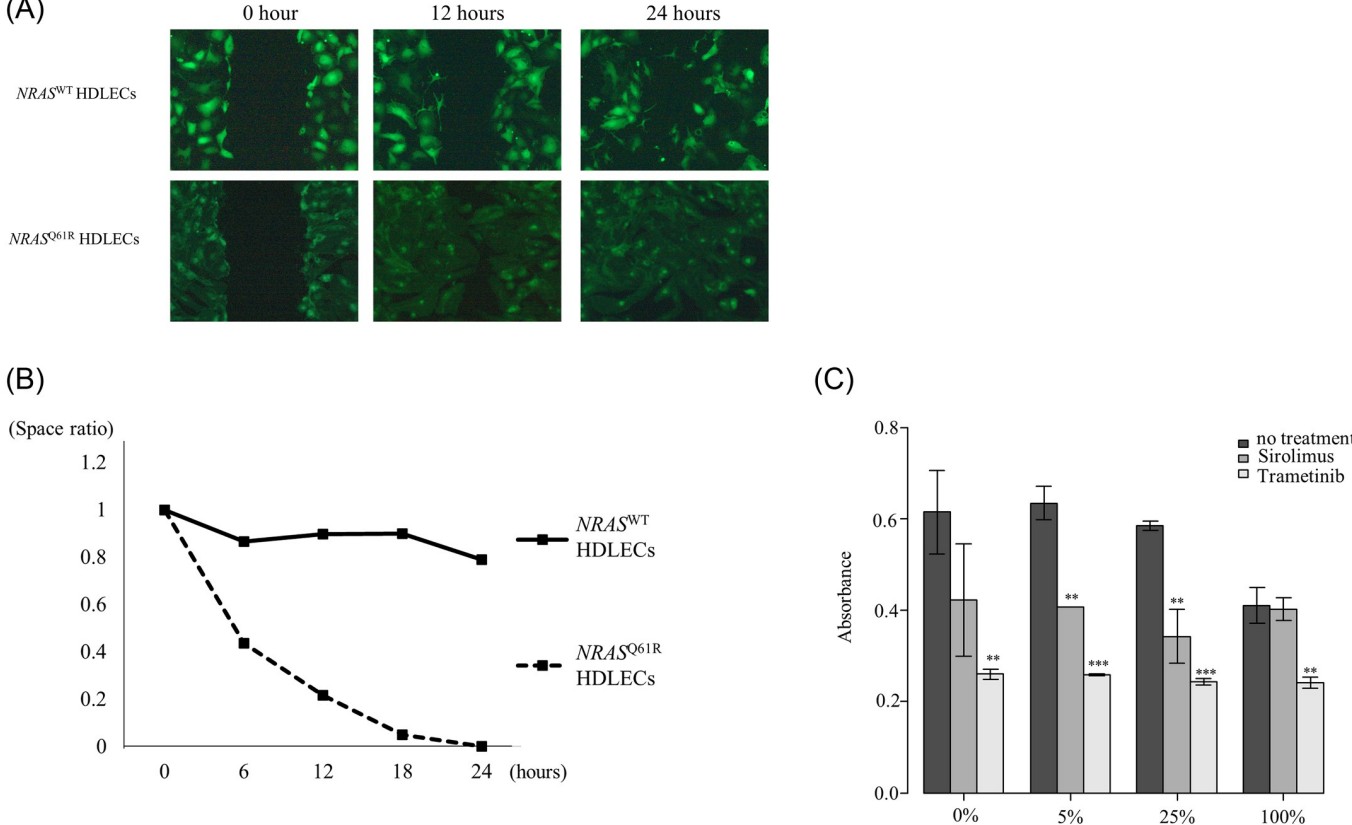

**Fig 3. Scratch assay and cell proliferation assay of *NRAS*$^{Q61R}$ and *NRAS*$^{WT}$ HDLECs.** 3A. Scratch assay of *NRAS*$^{Q61R}$ and *NRAS*$^{WT}$ HDLECs at 0, 6, 12, 18, and 24 h. Migration of *NRAS*$^{Q61R}$ HDLECs was more active than that of *NRAS*$^{WT}$ HDLECs. 3B. The space ratio of the wound area at 0, 6, 12, 18, and 24 h. The space ratio of *NRAS*$^{Q61R}$ HDLECs at 0, 6, 12, 18, and 24 h decreased more rapidly than that of *NRAS*$^{WT}$ HDLECs. Fig 3C. Absorbance of *NRAS*$^{Q61R}$ HDLEC mixtures treated with sirolimus and trametinib. Bars represent the mean ± SD of triplicate wells. *p<0.05, **p<0.01, ***p<0.001, compared with no treatment.

## Hyperactivation of the RAS pathway in *NRAS*<sup>Q61R</sup> HDLECs is improved by a MEK inhibitor

To determine the effects of the *NRAS* mutation on the PI3K/AKT/mTOR and RAS/MAPK/ERK pathways, AKT, p-AKT, ERK1/2, and p-ERK1/2 expressions were measured by western blot analysis. The expression levels of AKT in *NRAS*<sup>Q61R</sup> were similar to those of *NRAS*<sup>WT</sup> (Fig 4A). However, the expression levels of p-AKT were slightly reduced in *NRAS*<sup>Q61R</sup>. ERK1/2 and p-ERK/1/2 levels in *NRAS*<sup>Q61R</sup> were significantly upregulated compared with *NRAS*<sup>WT</sup>.

Additionally, to quantitatively measure protein expression levels, we performed a multiplex protein assay. The median fluorescence intensity (MFI) of IRS-1 decreased significantly, even in small amounts of *NRAS*<sup>Q61R</sup> HDLEC mixtures ($p<0.001$, $p<0.001$, and $p<0.001$, respectively) (S1a Fig). The MFI of AKT, the major protein of the PIK3 pathway, was decreased in the cell group containing 5% and 25% *NRAS*<sup>Q61R</sup> ($p = 0.021$ and $p = 0.007$, respectively), but there was no statistically significant difference between that of *NRAS*<sup>WT</sup> and *NRAS*<sup>Q61R</sup> (Fig 4B-a). The MFI of phosphorylated mTOR protein (p-mTOR) did not change regardless of the ratio of *NRAS*<sup>Q61R</sup> (Fig 4B-b). The MFI of p70S6K, a protein downstream of mTORC1, in *NRAS*<sup>Q61R</sup> HDLEC mixtures decreased significantly compared with that of *NRAS*<sup>WT</sup> ($p<0.001$, $p<0.001$, and $p = 0.002$, respectively) (S1b Fig). The MFI of phosphorylated ERK (p-ERK) was significantly increased in *NRAS*<sup>Q61R</sup> compared with *NRAS*<sup>WT</sup> ($p<0.001$) (Fig 4B-c). To assess the effects of sirolimus and trametinib treatment, we examined the expressions of AKT, p-mTOR, and p-ERK. The MFI of AKT after sirolimus and trametinib treatment was significantly decreased compared with that of no treatment ($p = 0.001$ and $p<0.001$, respectively) (Fig 4C-a). The MFI of p-mTOR after trametinib treatment was significantly decreased compared with that of no treatment ($p<0.001$) (Fig 4C-b). The MFI of p-ERK was significantly decreased after trametinib treatment ($p = 0.065$), but was not decreased after sirolimus treatment (Fig 4C-c).

## Increased secretion of ANG-2 and decreased secretion of VEGFR3 from *NRAS*<sup>Q61R</sup> HDLECs

Angiopoietin (ANG)-2, vascular endothelial growth factor (VEGF)-C, vascular endothelial growth factor receptor (VEGFR)-1, heparin binding-epidermal growth factor-like growth factor (HB-EGF), VEGFR-2, interleukin (IL)-8, and europilin-1 levels in the medium of *NRAS*<sup>WT</sup> and *NRAS*<sup>Q61R</sup> HDLECs were significantly higher than those of the controls (Fig 5a–5d; S2a–S2c Fig). ANG-2, VEGFR-1, HB-EGF, and neuroplin-1 expressions in *NRAS*<sup>Q61R</sup> were significantly higher than in *NRAS*<sup>WT</sup> HDLECs (Fig 5a, 5c and 5d; S2c Fig). VEGF-C and VEGFR-2 levels in *NRAS*<sup>Q61R</sup> were significantly lower than in *NRAS*<sup>WT</sup> (Fig 5b, S2a Fig). There was no significant difference in IL-8 levels between *NRAS*<sup>WT</sup> and *NRAS*<sup>Q61R</sup> (S2b Fig). G-CSF, SCF/c-kit, and TIE2 expressions were similar in *NRAS*<sup>WT</sup> and controls, and significantly higher in *NRAS*<sup>Q61R</sup> compared with *NRAS*<sup>WT</sup> and controls (Fig 5e, S2d and S2e Fig). The VEGF-A levels of *NRAS*<sup>WT</sup> and *NRAS*<sup>Q61R</sup> were lower than those of the controls (S2f Fig).

## The metabolome is altered in *NRAS*<sup>Q61R</sup> HDLECs

The number of confluent cells in 10 cm dishes was $1.3 \times 10^6$ cells for *NRAS*<sup>WT</sup> HDLECs and $0.4 \times 10^6$ cells for *NRAS*<sup>Q61R</sup> HDLECs. As mentioned above, there was a large difference in the size of the two cell lines; therefore, confluent cells in the 10 cm dishes were analyzed. The total amount of amino acids was similar in the two lines, with $213,587.0$ pmol/$1.3 \times 10^6$ cells in *NRAS*<sup>WT</sup> and $234,899.6/0.4 \times 10^6$ cells in *NRAS*<sup>Q61R</sup> HDLECs.

A bit

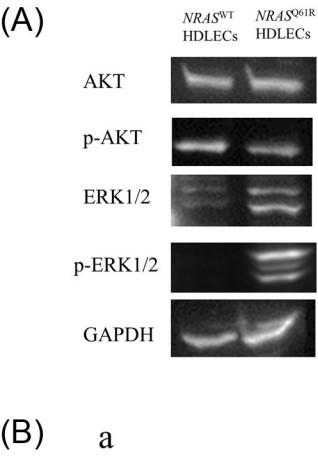

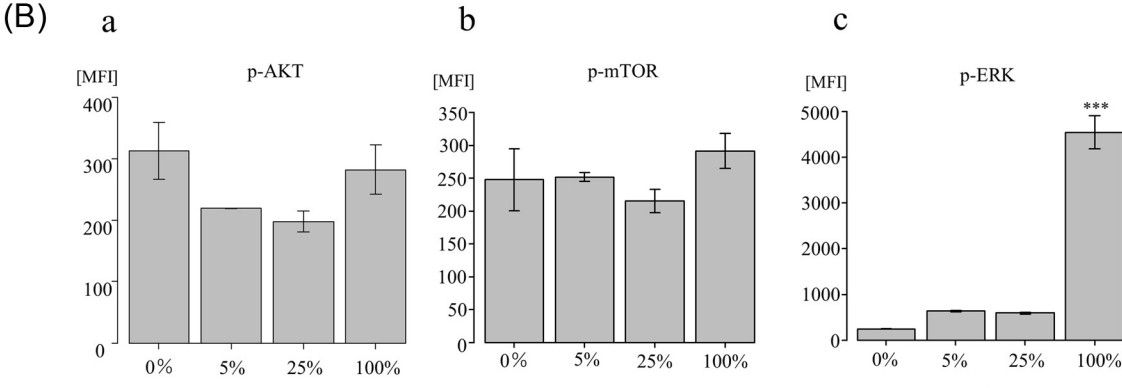

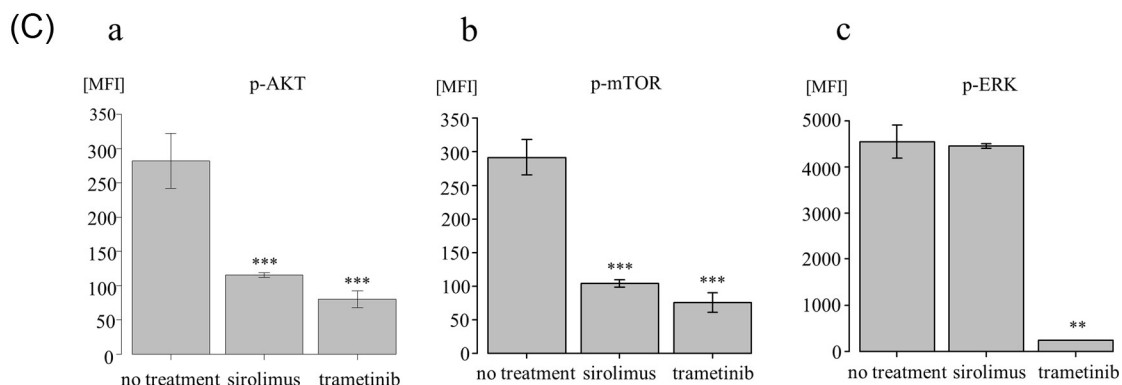

**Fig 4. Analysis of signaling pathways in HDLECs by western blotting and multiplex protein assay.** 4A. Western blot analysis of AKT, phospho-AKT (Ser473) (p-AKT [Ser473]), ERK1/2, and phospho-ERK1/2 (p-ERK1/2) expressions in *NRAS*^Q61R and *NRAS*^WT HDLECs. 4B. Multiplex protein assay of p-AKT (a), p-mTOR (b), p-ERK (c), in *NRAS*^Q61R and *NRAS*^WT HDLEC mixtures. Bars represent the mean ± SD of triplicate wells. *p<0.05, **p<0.01, ***p<0.001, compared with 0%. 4C. Multiplex protein assay of AKT (a), p-mTOR (b), and p-ERK(c) in *NRAS*^Q61R HDLECs with or without treatment with sirolimus and trametinib. Bars represent the mean ± SD of triplicate wells. *p<0.05, **p<0.01, ***p<0.001, compared with no treatment. After sirolimus and trametinib treatment, the absorbance of *NRAS*^Q61R HDLECs treated with sirolimus and trametinib decreased significantly compared with that of cells with no treatment.

Of the tricarboxylic acid (TCA) cycle factors examined, citric acid, cis-aconitic acid, isocitric acid, and α-ketoglutaric acid had significantly higher values in *NRAS*^Q61R than in *NRAS*^WT HDLECs (p = 0.007, p = 0.016, p = 0.016, and p = 0.006, respectively) (Fig 6). There was no significant difference in succinic acid and fumaric acid, and malic acid was significantly higher in

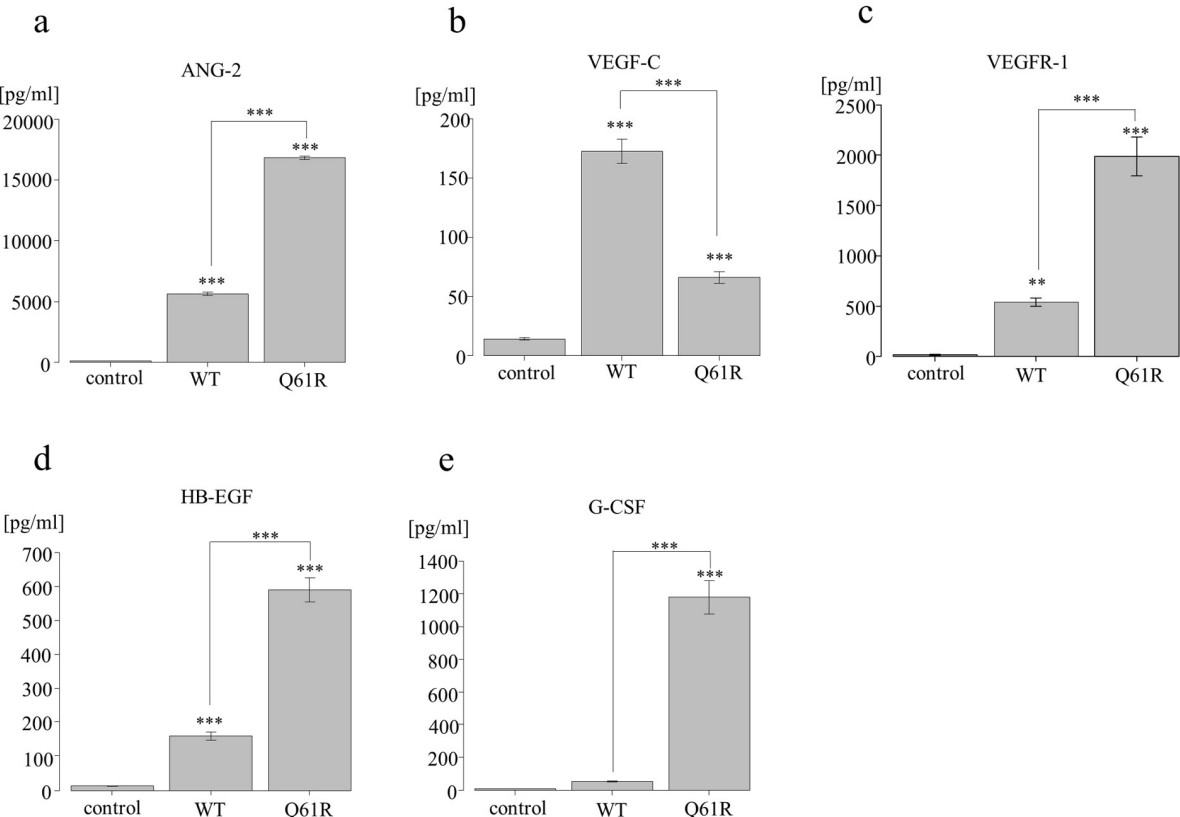

**Fig 5. Suspension array of *NRAS*^Q61R^ and *NRAS*^WT^ HDLECs.** Angiopoietin-2 (ANG-2) (a), vascular endothelial growth factor (VEGF)-C (b), vascular endothelial growth factor receptor (VEGFR)-1 (c), heparin binding-epidermal growth factor-like growth factor (HB-EGF) (d), and granulocyte-colony stimulating factor (G-CSF) (e) in *NRAS*^Q61R^ and *NRAS*^WT^ HDLECs. Bars represent the mean ± SD of triplicate wells. *p<0.05, **p<0.01, ***p<0.001, compared with controls. *NRAS*^Q61R^ and *NRAS*^WT^ HDLECs were compared.

*NRAS*^WT^ than in *NRAS*^Q61R^ (p = 0.037). 2-hydroxyglutaric acid synthesized from α-ketoglutaric acid was also not significantly different between *NRAS*^Q61R^ and *NRAS*^WT^.

Among other amino acids, aspartic acid in *NRAS*^WT^ HDLECs was significantly higher than in *NRAS*^Q61R^ (p = 0.008) (Fig 6). Asparagine and glutamic acid (Glu) in *NRAS*^Q61R^ were slightly higher than in *NRAS*^WT^, but there was no significant difference. Glutamine in both cell lines was not significantly different.

## Discussion

We report that *NRAS*^Q61R^ HDLECs had a higher migration ability and lower cell proliferation ability than *NRAS*^WT^ HDLECs. Tube formation deteriorated with increasing proportions of *NRAS*^Q61R^ HDLECs. Activation of the PIK3/AKT/mTOR pathway in *NRAS*^Q61R^ HDLECs was suppressed, and poor tube formation was improved by mTOR inhibitor treatment. These results indicate the theoretical *in vitro* circumstances induced by *NRAS*^Q61R^ mutated cells in the affected lesions of KLA patients.

The pathological findings of affected lesions in KLA patients showed morphological abnormalities in lymphatic cells and clusters of spindle cells. Although a very low frequency of the *NRAS* p.Q61R variant (1%–28%) was detected in samples of patients with KLA in previous studies [4], it is unknown which types of cells have this variant. In this study, we transfected

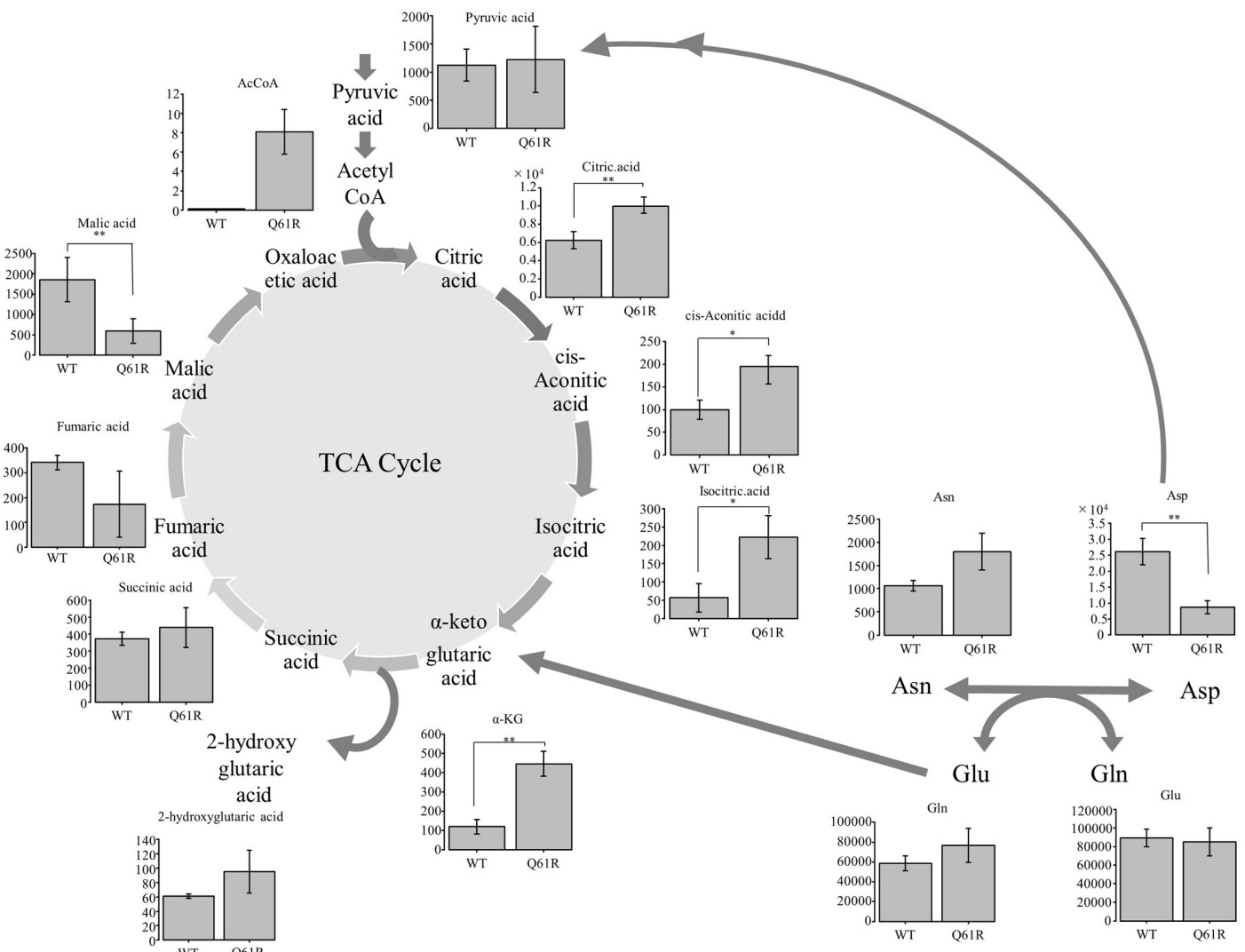

**Fig 6. Metabolome analysis of the TCA cycle in *NRAS*^Q61R and *NRAS*^WT HDLECs.** Bars represent the mean ± SD of triplicate wells. *p<0.05, **p<0.01, ***p<0.001.

*NRAS*^Q61R vectors into HDLECs, which had abnormal tube formation and proliferated in sheet form. In the mixed cells group, the higher the proportion of mutated cells, the larger the sheet-like area. Abnormal lymphatic tissue proliferation in KLA patients may also be caused by the partial loss of normal lymphatic construction.

A previous study examined the characteristics of cells isolated from two KLA patients, one of whom had the *NRAS*^Q61R mutation [7]. A tube formation assay showed that KLA cells had significantly fewer tubes than normal HDLECs. Their tubes proliferated in sheet form, similar to our *NRAS*^Q61R HDLECs [7]. Although it remains unclear why a small number of mutated HDLECs can affect the progress of lesions in KLA patients, our results using mixed *NRAS*^Q61R HDLECs suggest the hypothetical circumstances involved.

Recent studies indicated that the PIK3/AKT/mTOR and RAS/MAPK/ERK pathways are important in the pathogenesis of KLA [4, 5]. Western blotting revealed that the levels of p-AKT in *NRAS*^Q61R HDLECs were downregulated, but those of ERK1/2 and p-ERK1/2 were

upregulated. In a multiplex protein assay, IRS-1 and p70S6K levels in *NRAS*[Q61R] HDLECs were downregulated, but those of p-ERK1/2 were upregulated. These results showed that inactivation of the PIK3/AKT/mTOR pathway and hyperactivation of the RAS/MAPK/ERK pathway were caused by the activating *NRAS*[Q61R] mutation. A recent study of human endothelial progenitor cells expressing the *NRAS*[Q61R] mutation reported the same effects on these pathways [19]. However, established cell lines from three KLA patients demonstrated various results, including the downregulation of p-ERK or upregulation of p-AKT [7, 20]. Although it is challenging to predict whether these pathways are activated or inactivated in affected lesions, these established cell lines may contain mutated and non-mutated cells in pathological structures that have abnormal functions in the affected lesions. It is not yet clear how mutated and non-mutated cells associate with each other.

Sirolimus is a first-line drug for refractory CLAs, and trametinib is an alternative drug candidate [21, 22]. In previous studies, these drugs inhibited the proliferation of patient-derived cells [7, 19]. The present study showed that the tube formation and cell proliferation of *NRAS*[Q61R] HDLECs improved after both sirolimus and trametinib treatment. Trametinib inhibits the PIK3/AKT/mTOR pathway because it is closely related to the RAS pathway [23]. However, the proliferation of 100% *NRAS*[Q61R] was inhibited by treatment with trametinib, but not sirolimus. In a multiplex protein assay, the MFI of AKT and p-mTOR following sirolimus treatment was significantly decreased compared with that of non-treated cells, but that of p-ERK was decreased only after trametinib treatment. Similarly, Boscolo et al. reported that an MEK inhibitor, but not an mTOR inhibitor, improved the cellular shape of *NRAS*[Q61R] mutant human endothelial cells [19]. Our study findings did not fully clarify why mTOR and MEK inhibitors improved the tube formation and proliferation of patient-derived cells and mixed *NRAS*[Q61R], as well as *NRAS*[WT] HDLECs, or why only the MEK inhibitor prevented the proliferation of 100% *NRAS*[Q61R] HDLECs and cell circularity. We suspect that these findings might indicate the presence of a complex and potentially compensatory signaling pathway. Therefore, comprehensive functional studies are necessary to elucidate these mechanisms.

ANG-2 and VEGF-C are critical factors related to lymphangiogenesis [24]. ANG-2, which binds to TIE2, promotes maturation and stabilization after the remodeling of LECs [24–26]. VEGF-C, which binds to VEGFR-3, has an important role in vasculogenesis, sprouting, migration, and proliferation of lymphatic cells [24]. In our study, the levels of these proteins in the medium of *NRAS*[WT] and *NRAS*[Q61R] HDLECs were higher than those of the controls. ANG-2 levels in the medium of *NRAS*[Q61R] were significantly higher, but VEGF-C levels were lower, than those of *NRAS*[WT]. A tube formation assay showed that *NRAS*[Q61R] HDLECs proliferated excessively and formed tubes poorly. Therefore, we hypothesized that the hypersecretion of ANG-2 from *NRAS*[Q61R] HDLECs might decrease VEGFR-3 levels, associated with the inhibition of the maturation of peripheral LECs. However, this speculation requires additional functional analyses to confirm this relationship. Furthermore, previous studies of the cytokine profiles of KLA patients reported ANG-2 levels were elevated in patient samples, suggesting that it might be an important biomarker of KLA [27, 28]. Although the association between these cytokines and diseases is unknown, ANG-2 secretion and activation of these signaling cascades might be associated with the pathogenesis of KLA.

Growth factors such as G-CSF and HB-EGF in *NRAS*[Q61R] HDLECs were significantly higher than those in *NRAS*[WT]. It is considered that growth factors and cytokines produced by cancer cells within the tumor microenvironment might activate G-CSF in tumor and stromal cells because activation of the RAS/RAF/MEK pathway resulted in enhanced G-CSF expression [29]. HB-EGF is also thought to have a critical role in cell proliferation and differentiation. In particular, soluble HB-EGF is a potent promoter of cell adhesion, cell motility, and angiogenesis [30]. However, although the relationship between these growth factors and the

pathogenesis of vascular anomalies has not been elucidated, growth factors secreted by some *NRAS*$^{Q61R}$ HDLECs in affected lesions might influence the proliferation of mutated cells and abnormalities of peripheral normal LECs.

Many reports have shown that RAS is associated with metabolic changes in various cancer cell types [31–35], and mTOR is involved in glucose and amino acid metabolism [23]. We comprehensively analyzed metabolites and compared *NRAS*$^{WT}$ HDLECs with *NRAS*$^{Q61R}$. In the first half of the TCA cycle, which involves citric acid, cis-aconitic acid, isocitric acid, and α-ketoglutaric acid, the metabolite levels were significantly higher in *NRAS*$^{Q61R}$. α-Ketoglutaric acid is produced from isocitric acid in the TCA cycle and from Glu. Glu produced from Gln becomes α-ketoglutaric acid and enters the TCA cycle. Glutaminolysis is a metabolic pathway that extracts energy from glutamine, and is unique to cancer cells. It was reported that glutaminolysis is caused by the activation of *KRAS* in various cancers [33]. Thus, glutaminolysis is considered the major energy supply pathway in RAS-driven cancer [31]. Although it is not clear how glutaminolysis is associated with the RAS/MAPK/ERK pathway, our metabolome analysis demonstrated that *NRAS* mutations led to changes in cellular energy metabolism.

Our study suggests that the abnormal proliferation and morphological anomalies of lymphatic tissues within systemic lesions of patients with KLA are caused by the *NRAS*$^{Q61R}$ mutation in LECs. Furthermore, our data suggest that this is triggered by low-frequency somatic mosaic mutations. However, there were several limitations to this study. The first is that, although our results suggest the possibility that mutated cells influence surrounding cells, further evidence is needed to confirm this. The relationship between growth factors and the etiology of vascular anomalies is not understood; however, growth factors secreted by some *NRAS*$^{Q61R}$ HDLECs within the lesion may affect the proliferation of mutated cells and abnormalities in peripheral normal LECs. To investigate this further, additional functional analysis experiments will be necessary. Moreover, single-cell RNA sequencing of KLA samples or co-culturing *NRAS* mutant stromal cells with *NRAS*$^{WT}$ HDLECs might be a valuable approach. Such analysis might unravel the complex molecular and genetic interactions induced by mutated cells within a heterogeneous cell population, offering deeper insights into the cellular mechanisms underpinning KLA. The second limitation concerns the pharmacological effects of trametinib. In Fig 3C, the number of live cells after trametinib treatment was the same for all samples, regardless of the percentage of *NRAS*$^{Q61R}$ cells present. This suggests that the presence of factors other than those involved in RAS/MAPK/ERK pathway activation affected the drug's efficacy. Treatment with trametinib, regardless of mutation status, may have exerted a therapeutic effect on all LECs. Further investigation is needed on these points. Finally, although cell density was carefully adjusted to be uniform in all wells during sample preparation, the appearance of the baseline cell density differed between wild-type and mutant cells in the time-lapse photography. This discrepancy is attributed to the inherent size differences between the cell types. The mutant cells are naturally larger than the wild-type cells, which might give the impression of a higher or uneven density, even with an equivalent number of cells. The perceived difference in density is likely due to cell size variation, and not an actual difference in cell numbers.

## Conclusions

*NRAS*$^{Q61R}$ HDLECs showed poor tube formation, low cell proliferation, and high migration ability, and an increasing ratio of mutated cells led to poorer tube formation. mTOR and MEK inhibitors improved tube formation and decreased cell proliferation. Mutations in the *RAS* genes of LECs drive the incessant activation of critical pathways, which in turn dysregulate the function of cytosolic and nuclear signaling effector molecules, such as cytokines and metabolic

regulatory proteins. This might lead to abnormal tissue functions and the differentiation of affected lesions in CLA patients.

## Supporting information

**S1 File. Time-lapse photography of tube formation by *NRAS*^WT^ HDLECs.**
(MP4)

**S2 File. Time-lapse photography of tube formation by *NRAS*^Q61R^ HDLECs.** *NRAS*^Q61R^ HDLECs showed impaired formation of uniform and continuous loops.
(MP4)

**S1 Fig. Multiplex protein assay of IRS-1 (a), p70S6K (b) in *NRAS*^Q61R^ and *NRAS*^WT^ HDLEC mixtures.** Bars represent mean ± SD from triplicate wells. *p<0.05, **p<0.01, ***p<0.001, compared with 0%.
(TIF)

**S2 Fig. Suspension array of *NRAS*^Q61R^ and *NRAS*^WT^ HDLECs.** Vascular endothelial growth factor receptor (VEGFR)-2 (a), interleukin (IL)-8 (b), neuroplin-1 (c), stem cell factor (SCF)/c-kit (d), TIE2 (e), vascular endothelial growth factor (VEGF)-A (f), VEGFR-3 (g), hepatocyte growth factor (HGF) (h), VEGF-D (i), tenascin-C (j), and leptin (k). Bars represent mean ± SD from triplicate wells. *p<0.05, **p<0.01, ***p<0.001, compared with control. *NRAS*^Q61R^ and *NRAS*^WT^ HDLECs were also compared.
(TIF)

**S1 Raw image.**
(PDF)

**S2 Raw image.**
(PDF)

**S3 Raw image.**
(PDF)

**S4 Raw image.**
(PDF)

**S5 Raw image.**
(PDF)

## Acknowledgments

We thank the Department of Pediatrics at Gifu University for their contribution. We also thank Ms. Asuka Ogawa of Gifu University for providing technical assistance. Finally, we thank H. Nikki March, PhD and J. Ludovic Croxford, PhD, from Edanz (https://jp.edanz.com/ac) for editing drafts of this manuscript.

## Author Contributions

**Conceptualization:** Michio Ozeki.

**Investigation:** Shiho Yasue, Akifumi Nozawa, Saori Endo.

**Supervision:** Michio Ozeki, Hidenori Ohnishi.

**Writing – original draft:** Shiho Yasue.

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
