## [Decision Letter · Decision Letter 0]

26 Oct 2023

PONE-D-23-21611Changes in cell morphology and function induced by NRAS Q61R mutation in lymphatic endothelial cellsPLOS ONE

Dear Dr. Yasue,

Thank you for submitting your manuscript to PLOS ONE. After careful consideration, we feel that it has merit but does not fully meet PLOS ONE’s publication criteria as it currently stands. Therefore, we invite you to submit a revised version of the manuscript that addresses the points raised during the review process.

We look forward to receiving your revised manuscript.

Kind regards,

Avaniyapuram Kannan Murugan, M.Phil., Ph.D.

Academic Editor

PLOS ONE

Journal Requirements:

The name of the colleague or the details of the professional service that edited your manuscript.A copy of your manuscript showing your changes by either highlighting them or using track changes (uploaded as a *supporting information* file).A clean copy of the edited manuscript (uploaded as the new *manuscript* file).

“..The present study was supported in part by a Practical Research Project for Rare/Intractable Diseases (JP22ek0109515) from Japan’s Agency for Medical Research and Development, AMED. Finally, we thank H. Nikki March, PhD, from Edanz (https://jp.edanz.com/ac) for editing a draft of this manuscript.”

“The authors received no specific funding for this work.”

**Additional Editor Comments:**

1. Though manuscript has merit, reviewers raised many concerns about the article and related experiments and is important to be addressed before consideration for publication. 

2. The following articles are potentially important in RAS related research and hence can be cited them in the revised version (PMID: 31255772; PMID: 27102293; PMID: 26620726).

Reviewers' comments:

Reviewer's Responses to Questions

**Comments to the Author**

1. Is the manuscript technically sound, and do the data support the conclusions?

Reviewer #1: Yes

Reviewer #2: Partly

Reviewer #3: Yes

2. Has the statistical analysis been performed appropriately and rigorously? 

Reviewer #1: Yes

Reviewer #2: I Don't Know

Reviewer #3: Yes

3. Have the authors made all data underlying the findings in their manuscript fully available?

Reviewer #1: Yes

Reviewer #2: Yes

Reviewer #3: Yes

4. Is the manuscript presented in an intelligible fashion and written in standard English?

Reviewer #1: Yes

Reviewer #2: Yes

Reviewer #3: Yes

5. Review Comments to the Author

Reviewer #1: The manuscript, “Changes in cell morphology and function induced by NRAS Q61R mutation in lymphatic endothelial cells” presents the morphological characterization and functional profiling of human dermal lymphocyte endothelial cells (LECs) upon transfection with lentivirus vector carrying specific NRAS mutation. The work includes protein expression analysis, and metabolomic profiling upon such transfection. According to this in vitro investigation NRAS Q61R human dermal LECs have poor tube formation and high cell proliferation and migration ability with increasing ratios of mutated cells. They demonstrate the inactivation of the PIK3/AKT/mTOR pathway and hyperactivation of the RAS/MAPK/ER pathway in the transfected cells. The latter is inhibited by trametinib, which targets MAP kinase. The study provides a possible mechanism for Kaposis lymphangiomatosis pathology and suggests the potential treatment strategies.

The study is well designed and presented in a coherent manner. There are a few presentation and language concerns:

Presentation Concerns:

i. In the Materials and Methods section under the heading “Gene transfection of HDLEC cell line using lentiviruses”, the authors have written “NRASQ61R expression in these cells was checked by polymerase chain reaction (PCR).” Probably they mean insertion rather than expression. In case of expression profiling “real time PCR” must have been performed. The authors may provide details as supplementary material.

ii. Figure 1: The magnification scale bar needs to be provided.

iii. Images (a) and (b) in Figure 1 need to be more clear to draw conclusions in the text.

iv. Under Metabolomics of HDLECs heading on page no. 26, last paragraph, Fig. 5. Is written instead of Fig. 6.

Language suggestions:

The manuscript may be revised for some minor language corrections for the purpose of clarity and precision. For instance:

i. In the abstract, “MEK” abbreviation is not explained.

ii. In the Introduction, it’s unclear what authors mean by “KLA is a subtype of GLA … [and presents with] poorer outcomes than GLA patients”. Do they actually mean “worse outcomes than GLA patients with other subtypes”?

iii. In the Methods section under the heading “Gene transfection of HDLEC cell line using lentiviruses”, the authors have written “The recombinant lentiviral solution was diluted with EBM2 to give solutions with initial concentrations of 1:4 to 1:10.” May be the authors should consider “prepare solutions with initial concentrations of 1:4 to 1:10”.

Reviewer #2: The manuscript by Yasue et al. describes the morphological and functional alterations observed in juvenile foreskin-derived human dermal lymphatic endothelial cells (HDLECs) after stable lentiviral insertion of a mutated form of the NRAS gene (Q61R variant). The Authors argue that somatic low-level mutation of the NRAS gene in HDLECs could explain the lymphatic abnormalities observed in patients with kaposiform lymphangiomatosis (KLA).

However, several concerns weaken the reliability of this work. The main concerns relate to the consistency and presentation of the data. Furthermore, the analysis presented is strictly descriptive, functional experiments are required to support the claimed correlation and causality. Previously published papers report the NRASQ61R variant found in KLA samples, with no evidence as to which cell type is mutated. The Authors propose that the NRASQ61R mutation of HDLECs is capable of inducing KLA-like aberrations based on similar molecular and morphological abnormalities observed in vitro. At the same time, they suggest that low-frequency mutated cells could affect the phenotype of distant cells even regardless of their mutational status, thus raising questions about the ability to induce the same phenotype on HDLECs by RAS mutation in other cell types in the microenvironment. To clarify this point, single cell RNA sequencing of KLA samples or coculture of WT HDLECs with NRAS mutant stromal cells should be performed.

MAJOR

- Intensive improvement of the image quality in the figures is required to appreciate the morphology of the cells and structures

- The WT cells in most of the figures are not visible as they are not colored. This is especially inconvenient when mixed cultures are being analyzed and NRASQ61R cells are the only observable, with no WT cell information in the dark areas. Red staining should be performed on WT cells to better understand the cell-cell interactions and morphological changes.

- The time-lapse photography in supplementary files a, b, allows observation of WT and mutant cells, however cell density at baseline appears to be different. Can you explain this discrepancy?

- The results of the proliferation assay should be better clarified: an increase in proliferation in NRASQ61R cells is reported in the abstract and discussion, while the opposite phenotype is described in the results, in figure 3C (decreased in NRASQ61R cells) and in the conclusion

- The same consideration for the number of nets formed in NRASQ61R cells with respect to WT: the result described in the text is opposite with respect to the graph of figure 2B. Please clarify this point.

- In figure 3C the amount of live cells after trametinib treatment is the same in all samples, regardless of the percentage of NRASQ61R cells; how can this apparent independence of drug efficacy from ERK MAPK activation be explained?

- Analysis of the metabolome of NRASQ61R cells revealed increased secretion of ANG-2 and decreased VEGFR3 compared with WT cells. The authors suggested that hypersecretion of ANG-2 by NRASQ61R HDLECs may cause decrease in VEGFR-3 levels, thereby imparing tube formation. Functional experiments based on loss or gain of function are needed to strenghten this claim.

MINOR

- The titles in the results section are more suitable for materials and methods; please replace them with short sentences that briefly describe the result obtained

- Figures 4B-d and 4B-e are mentioned in the text but missing from the figure

- Please specify in figure 3B what it refers to: the legend of the figure refers to the proliferation data, while the text shows the space ratio result and the title on the y axis is missing

- in the last paragraph of the results chapter “Analysis of signaling pathways in HDLECs by western blotting and multiplex protein assay” there are some refuses to correct: AKT was described in the previous paragraph, so it shoud be something different; moreover, only the efficacy of trametinib is described while two P-values are reported which raise doubts about the described target.

- Please check the captions in the graphs of Figures 4B and 4C, they look confusing

- In the M&M section, the chapter “Analysis of NRAS mutational status” appears to be a refuse of the previous paper by the group, as NRAS mutational status was not analyzed in this study

Reviewer #3: In this manuscript, Yasue et al, aimed to explore a low-level somatic mutation in NRAS gene (c.182 A > G, Q61R) in LA patients. The authors generated stable lymphatic endothelial cells (LECs) with expressing NRASQ61R mutation using Lentiviral system to understand morphological and functional characterization, protein expression profiling, and metabolome analysis of those mutated cells compared to WT cells. They found out that NRASQ61R human dermal LECs showed poor tube formation and high cell proliferation and migration ability with increasing ratios of mutated cells. Moreover, signaling pathways analysis showed inactivation of the PIK3/AKT/mTOR pathway and hyperactivation of the RAS/MAPK/ERK pathway, which was improved by MEK inhibitor treatment. Overall, the findings they present are interesting, technically sound, and could be valuable to the field. Their claims are convincing and supported by the experimental data with comprehensive experimental controls. The manuscript provides relevant, up-to-date information and it is well written and supported with proper figures. All the data appears to be available within the manuscript and supplementary data.

I have minor suggestions that may be helpful for the reader to better understand the current study.

1. Figure 1: NRASQ61R HDLECs showed larger and irregular-shaped morphology compared with NRASWT HDLECs microscopically (Fig.1 a–c). Please define the image name which one is WT or Q61R? If there is no WT cell images, please include it as well.

2. Figure 2B: It is better to include X-axis name as (e.g. WT: Q61R cell ratio) in the figure for better understanding

3. Figure 2D: It is better to include “DMSO treated control cells” to show the changing of the poor tube-forming ability.

4. Figure 3B: Please define the Y-axis name in the graph (e.g. space ratio)

5. Figure 5: What is the control group?

6. Page 26: “Among other amino acids, aspartic acid in NRASWT HDLECs was significantly higher than in NRASQ61R (p=0.008) (Fig.5)” Figure number should be 6

6. PLOS authors have the option to publish the peer review history of their article (what does this mean?). If published, this will include your full peer review and any attached files.

Reviewer #1: **Yes**

Reviewer #2: **Yes**

Reviewer #3: No

---

## [Author Response · Author response to Decision Letter 0]

10 Jan 2024

PONE-D-23-21611

Changes in cell morphology and function induced by NRAS Q61R mutation in lymphatic endothelial cells

PLOS ONE

Dear Avaniyapuram Kannan Muruganv M.Phil., Ph.D.

Academic Editor

PLOS ONE

Thank you for advise for submitting this article.

We wrote in response to the recent feedback regarding my manuscript titled "Changes in cell morphology and function induced by NRAS Q61R mutation in lymphatic endothelial cells," which I had the honor of submitting to PLOS ONE.

 I am truly grateful for the opportunity to revise my manuscript based on the insightful comments received during the review process.

The amended Funding Statement is as follows: "The present study was supported in part by a Practical Research Project for Rare/Intractable Diseases (JP22ek0109515) from Japan’s Agency for Medical Research and Development, AMED." We trust that this meets the requirements for the Funding Statement section and appreciate your assistance in updating our submission records. Please change the online submission form on my behalf.

I added the source of funding that contributed to my study in this form.　Please comfirm it.

Sincerly, 

Shiho Yasue

---

## [Decision Letter · Decision Letter 1]

7 Feb 2024

PONE-D-23-21611R1Changes in cell morphology and function induced by the NRAS Q61R mutation in lymphatic endothelial cellsPLOS ONE

Dear Dr. Ozeki,

Thank you for submitting your manuscript to PLOS ONE. After careful consideration, we feel that it has merit but does not fully meet PLOS ONE’s publication criteria as it currently stands. Therefore, we invite you to submit a revised version of the manuscript that addresses the points raised during the review process.

We look forward to receiving your revised manuscript.

Kind regards,

Avaniyapuram Kannan Murugan, M.Phil., Ph.D.

Academic Editor

PLOS ONE

Journal Requirements:

Additional Editor Comments:

Authors have partially addressed the comments: The following points to be taken care before considering the article for publication.

Comments;

1. The Reviewer #2 still has some points which is critical.

2. The following important RAS-related articles to be cited in the manuscript.

Importance of RAS in human cancer: PMID: 31255772; PMID: 27102293

How RAS can be modulated by miRNAs: PMID: 26620726

RAS mutations in other cancers: PMID: 22240207; PMID: 19628422

3. The spell, grammar and expressions to be checked as PLoS one does not edit for Grammar.

Reviewers' comments:

Reviewer's Responses to Questions

**Comments to the Author**

1. If the authors have adequately addressed your comments raised in a previous round of review and you feel that this manuscript is now acceptable for publication, you may indicate that here to bypass the “Comments to the Author” section, enter your conflict of interest statement in the “Confidential to Editor” section, and submit your "Accept" recommendation.

Reviewer #2: (No Response)

Reviewer #3: All comments have been addressed

2. Is the manuscript technically sound, and do the data support the conclusions?

Reviewer #2: Partly

Reviewer #3: Yes

3. Has the statistical analysis been performed appropriately and rigorously? 

Reviewer #2: I Don't Know

Reviewer #3: Yes

4. Have the authors made all data underlying the findings in their manuscript fully available?

Reviewer #2: Yes

Reviewer #3: Yes

5. Is the manuscript presented in an intelligible fashion and written in standard English?

Reviewer #2: Yes

Reviewer #3: Yes

6. Review Comments to the Author

Reviewer #2: The Authors have provided a revised version of the manuscript entitled “Changes in cell morphology and function induced by the NRAS Q61R mutation in lymphatic endothelial cells”. Although the Authors have addressed all the minor concerns and most of the major concerns that have been raised, some unaddressed points still persist:

- The mixed cultures in figure 1c are still difficult to evaluate as only NRASQ61R cells are visible. Since enhanced images of the NRASWT and NRASQ61R HDLECs with fluorescence have been added, consider deleting figure 1c.

- Since no functional experiments have been performed demonstrating that hypersecretion of ANG-2 by NRASQ61R HDLECs leads to decreased VEGFR-3 levels, the work is still descriptive and provides only speculative conclusions.

- the same perplexity concerns the effect of inhibitors of the MEK and mTOR pathways which is not clearly linked to the NRAS mutation. It is unclear why mTOR inhibition has the same effects as MEK inhibition in cells without hyperactivation of the PI3K/AKT/mTOR pathway.

Reviewer #3: All my comments have been addressed by the author. I have no more suggestions. The paper is ready for publishing.

7. PLOS authors have the option to publish the peer review history of their article (what does this mean?). If published, this will include your full peer review and any attached files.

Reviewer #2: No

Reviewer #3: No

---

## [Author Response · Author response to Decision Letter 1]

3 May 2024

PONE-D-23-21611

Title: Changes in cell morphology and function induced by the NRAS Q61R mutation in lymphatic endothelial cells

Journal Requirements:

We will review our reference list again to ensure its completeness and accuracy. We confirm that there are no retracted papers cited in our manuscript. Thank you for your guidance on this matter.

Additional Editor Comments:

Authors have partially addressed the comments: The following points to be taken care before considering the article for publication.

We thank the editor and reviewers for their helpful comments on our manuscript, which have contributed to its improvement. Our responses are outlined below. Changes to the text are in red font.

Comments;

1. The Reviewer #2 still has some points which is critical.

We have addressed the critical points raised by Reviewer #2 as follows. Thank you for bringing them to our attention.

2. The following important RAS-related articles to be cited in the manuscript.

Importance of RAS in human cancer: PMID: 31255772; PMID: 27102293

How RAS can be modulated by miRNAs: PMID: 26620726

RAS mutations in other cancers: PMID: 22240207; PMID: 19628422

Thank you for recommending these invaluable references on the importance of RAS in human cancer, its modulation by miRNAs, and RAS mutations in other cancers. We are grateful for your guidance and have duly cited these articles in our manuscript to enrich our discussion and support our findings. Your suggestions have significantly contributed to the depth and accuracy of our work.

“34. Murugan AK, Grieco M, Tsuchida N. RAS mutations in human cancers: Roles in precision medicine. Semin Cancer Biol. 2019 Dec;59:23-35. doi: 10.1016/j.semcancer.2019.06.007.

35. Tsuchida N, Murugan AK, Grieco M. Kirsten Ras* oncogene: significance of its discovery in human cancer research. Oncotarget. 2016 Jul 19;7(29):46717-46733. doi: 10.18632/oncotarget.8773.” (P34, L596- P35, L600)

3. The spell, grammar and expressions to be checked as PLoS one does not edit for Grammar.

Thank you for the reminder. We have reviewed the manuscript and corrected spelling, grammar, and expressions where necessary.

Responses to Reviewer 2

The Authors have provided a revised version of the manuscript entitled “Changes in cell morphology and function induced by the NRAS Q61R mutation in lymphatic endothelial cells”. Although the Authors have addressed all the minor concerns and most of the major concerns that have been raised, some unaddressed points still persist:

- The mixed cultures in figure 1c are still difficult to evaluate as only NRASQ61R cells are visible. Since enhanced images of the NRASWT and NRASQ61R HDLECs with fluorescence have been added, consider deleting figure 1c.

Thank you for your insightful comments and for highlighting the issues with Figure 1c in our manuscript. We appreciate your detailed review and agree with your assessment that the mixed cultures presented in Figure 1c are difficult to evaluate, because only NRASQ61R cells are prominently visible. As you suggested, we have removed Figure 1c from our manuscript.

We have made the necessary revisions to the text to reflect the removal of this figure and ensure that the manuscript remains cohesive and that the flow of information is not disrupted. We hope that this modification addresses your concern and improves the quality and readability of our manuscript.

- Since no functional experiments have been performed demonstrating that hypersecretion of ANG-2 by NRASQ61R HDLECs leads to decreased VEGFR-3 levels, the work is still descriptive and provides only speculative conclusions.

Thank you for your thoughtful feedback and for drawing attention to a crucial aspect of our study. We acknowledge the limitation you highlighted regarding the lack of direct functional experiments to demonstrate the impact of ANG-2 hypersecretion by NRASQ61R HDLECs on VEGFR-3 levels. As you correctly pointed out, our study of the relationship between ANG-2 hypersecretion and VEGFR-3 levels, as well as its subsequent effect on the maturation of peripheral LECs, is speculative and based on correlative evidence. Our statement was intended to propose a potential mechanistic insight based on the observed phenomena. In light of your feedback, we agree that it is imperative to clarify the speculative nature of these conclusions. Therefore, we revised our manuscript to include an explicit statement acknowledging the absence of direct functional experiments.

“Therefore, we hypothesized that the hypersecretion of ANG-2 from NRASQ61R HDLECs might decrease VEGFR-3 levels, associated with the inhibition of the maturation of peripheral LECs. However, this speculation requires additional functional analyses to confirm this relationship.” (P25, L427-429)

- the same perplexity concerns the effect of inhibitors of the MEK and mTOR pathways which is not clearly linked to the NRAS mutation. It is unclear why mTOR inhibition has the same effects as MEK inhibition in cells without hyperactivation of the PI3K/AKT/mTOR pathway.

Thank you for your insightful observation and for highlighting the complex dynamics between the NRAS mutation and effects of MEK and mTOR inhibitors. We acknowledge the significance of your concern and agree that the mechanisms underlying these observations are not fully understood at this stage of our research.

In our study, we observed that inhibitors targeting the MEK and mTOR pathways exhibited similar effects, even in cells where the PI3K/AKT/mTOR pathway was not hyperactivated. This finding raises important questions about the specificity and broader implications of these inhibitors in the context of NRAS mutations. We speculate that the convergence of the inhibitory effects of these pathways, despite the lack of overt hyperactivation of the PI3K/AKT/mTOR pathway, suggests a more complex, possibly compensatory, network of signaling interactions. However, we must emphasize that this interpretation is preliminary and necessitates further investigation. To address this gap in our understanding, we have added a section to our discussion highlighting the perplexity surrounding these effects and acknowledging the need for comprehensive functional studies. We think that elucidating these mechanisms will be crucial for developing more targeted and effective therapeutic strategies.

We have revised the text with reference to the points above.

“Our study findings did not fully clarify why mTOR and MEK inhibitors improved the tube formation and proliferation of patient-derived cells and mixed NRASQ61R, as well as NRASWT HDLECs, or why only the MEK inhibitor prevented the proliferation of 100% NRASQ61R HDLECs and cell circularity. We suspect that these findings might indicate the presence of a complex and potentially compensatory signaling pathway. Therefore, comprehensive functional studies are necessary to elucidate these mechanisms.” (P24, L414-419)

We appreciate your highlighting of this issue, which underscores the complexity of cell signaling and the challenges in targeting these pathways. Your comment has been instrumental in guiding us to refine our manuscript and outline the directions for future research. Once again, we thank you for your constructive feedback and hope that our revisions and acknowledgment of these limitations reflect a thoughtful consideration of the complexities involved.

Responses to Reviewer 3

Reviewer #3: All my comments have been addressed by the author. I have no more suggestions. The paper is ready for publishing.

Thank you for your constructive feedback and for confirming that all your comments have been addressed satisfactorily. We appreciate your thorough review and are grateful for your support in deeming our paper ready for publication.

---

## [Editor Report · Decision Letter 2]

7 May 2024

Changes in cell morphology and function induced by the NRAS Q61R mutation in lymphatic endothelial cells

PONE-D-23-21611R2

Dear Dr. Ozeki,

We’re pleased to inform you that your manuscript has been judged scientifically suitable for publication and will be formally accepted for publication once it meets all outstanding technical requirements.

Kind regards,

Avaniyapuram Kannan Murugan, M.Phil., Ph.D.

Academic Editor

PLOS ONE
---

## [Editor Report · Acceptance letter]

16 May 2024

PONE-D-23-21611R2 

PLOS ONE

Dear Dr. Ozeki, 

I'm pleased to inform you that your manuscript has been deemed suitable for publication in PLOS ONE. Congratulations! Your manuscript is now being handed over to our production team.

Kind regards, 

on behalf of

Dr. Avaniyapuram Kannan Murugan 

Academic Editor

PLOS ONE